# Safety of the bag-in-the-lens implantation regarding the development of clinically significant pseudophakic cystoid macular edema: A retrospective case series study

Dorothée Scheers[1,2©], Jasmien Rens[1,2©]*, Luc Van Os[1,2], Sorcha Ní Dhubhghaill[1,2], Veva De Groot[1,2,3], Stefan Kiekens[1], Jan Van Looveren[2], Kristien Wouters[2,4], Marie-José Tassignon[1,2]

1 Department of Ophthalmology, University Hospital Antwerp, Edegem, Belgium, 2 Faculty of Medicine and Health Sciences, Department of Ophthalmology, Visual Optics and Visual Rehabilitation, University of Antwerp, Edegem, Belgium, 3 Department of Ophthalmology, Middleheim Hospital, Antwerp, Belgium, 4 Clinical Trial Center (CTC), CRC Antwerp, University Hospital Antwerp, University of Antwerp, Edegem, Belgium

© These authors contributed equally to this work.
* jasmien.rens@uza.be

## Abstract

### Purpose

To determine the incidence of clinically significant pseudophakic cystoid macular edema (CSPME) after phacoemulsification using the 'bag-in-the-lens' lens (BIL) implantation technique and to examine the influence of associated risk factors for clinically significant pseudophakic macular edema (CSPME), both ocular and systemic.

### Methods

This retrospective study included 2419 first-operated eyes of 2419 adults who underwent phacoemulsification cataract surgery using the BIL implantation technique between January 2013 and December 2018 in the Antwerp University Hospital, Belgium. The significance of several risk factors (age, gender, previous history, intra- and postoperative complications) was examined by extraction of electronic medical files.

### Results

The 3-month incidence of CSPME in the subgroup without risk factors was 0.00% (95% CI: 0.00 –NA). The 3-month incidence of CSPME in the subgroup with risk factors was 0.57% (95% CI 0.22–1.29%).

The 3-month incidence of CSPME in the total population of 2419 patients was 0.29% (95% CI: 0.11–0.65%). The risk factors most significantly associated with CSPME included renal insufficiency (hazard ration [HR]: 5.42; 95% CI: 1.69–17.44; P = .014), exudative age-related macular degeneration (HR: 74.50, 95% CI: 25.75–215.6; P < .001) and retinal vein occlusion (HR: 22.48, 95% CI: 4.55–111.02; P = .005).

**Data Availability Statement:** All relevant data are within the paper and its Supporting Information file.

**Funding:** The funders had no role in study design, data collection and analysis, decision to publish, or preparation of the manuscript.

**Competing interests:** Professor Tassignon has intellectual property rights to the bag-in-the-lens IOL and the ring caliper, which is licensed to Morcher GmbH, Stuttgart, Germany. There are no patents, products in development or marketed products associated with this research to declare. This does not alter our adherence to PLOS ONE policies on sharing data and materials.

## Conclusions

In the absence of risk factors, the incidence of CSPME was zero. We can conclude that Primary Posterior Continuous Curvilinear Capsulorhexis (PPCCC) does not increase the risk for CSPME.

Non-inferiority of the BIL implantation regarding the development of CSPME, relative to the traditional 'lens-in-the-bag' (LIB) implantation, confirms that BIL is a safe surgical technique. This study also illustrates a previously undescribed risk factor for developing CSPME, namely renal insufficiency.

## Introduction

The implantation of an intraocular lens (IOL), after the removal of the natural crystalline lens, is the standard in modern cataract surgery. The most common method is to place the IOL into the polished capsular bag, once all residual cortical material has been removed. The "lens-in-the-bag" (LIB) placement secures the IOL in a position similar to the natural lens. While the capsular bag approach is safe, it is still associated with some complications. Postoperative capsular problems like posterior capsule opacification (PCO) and capsular contraction can occur. PCO is a relatively frequent post-surgical complication and is caused by the proliferation of equatorial lens epithelial cells left behind in the capsular bag. The incidence of PCO ranges from 5% in patients with no risk factors and uncomplicated cataract surgery to 50% in patients with risk factors, such as uveitis, diabetes, pediatric or traumatic cataracts, or complicated surgeries. The longer the postoperative time, the higher the incidence of PCO [1–5].

The bag-in-the-lens (BIL) implantation technique is an approach that avoids this complication. The BIL IOL is a monofocal spherical (or toric) hydrophilic IOL with a biconvex optic and 2 elliptical plane haptics (Fig 1) [6]. The lens implantation differs from the traditional approach because it requires a primary posterior continuous curvilinear capsulorhexis (PPCCC) of the same size as the anterior capsulorhexis [5,7]. The IOL is then suspended by the capsular bag after the anterior and posterior lens capsules are placed into the equatorial groove of the IOL, thus placing the bag in the lens. The BIL IOL sequesters any residual lens epithelial cells into a structurally sealed space, and as a result, PCO does not occur [8–10]. This property is particularly interesting in patients susceptive to PCO, e.g.: children, patients with diabetes, and patients with uveitis [5,11–13].

While the BIL implantation technique confers advantages, it does also require additional surgical maneuvers which need to be mastered by a surgeon choosing to implant the BIL. It can therefore be asked whether this additional complexity is accompanied by additional risks [14,22]. Traumatic posterior capsular rupture is known be associated with an increased risk of vitreous prolapse and pseudophakic cystoid macular edema (PCME). A planned PPCCC however is quite different to an inadvertent capsular rent because care is taken to push the anterior hyaloid posteriorly using an ophthalmic viscosurgical device (OVD) prior to completing the PPCCC, leaving the anterior hyaloid face untouched [11]. Maintaining the anterior hyaloid face intact appears to be the fundamental feature to prevent problems as associated with unintended posterior capsular rupture [6,14,15].

PCME, (Irvine-Gass syndrome), originally reported as a consequence of intracapsular cataract extraction, is a sight-threating postoperative complication though its rate has been decreasing over the past decades [16–18]. This reduction is due to improved postoperative care protocols, smaller incisions, and introduction of extracapsular techniques such as

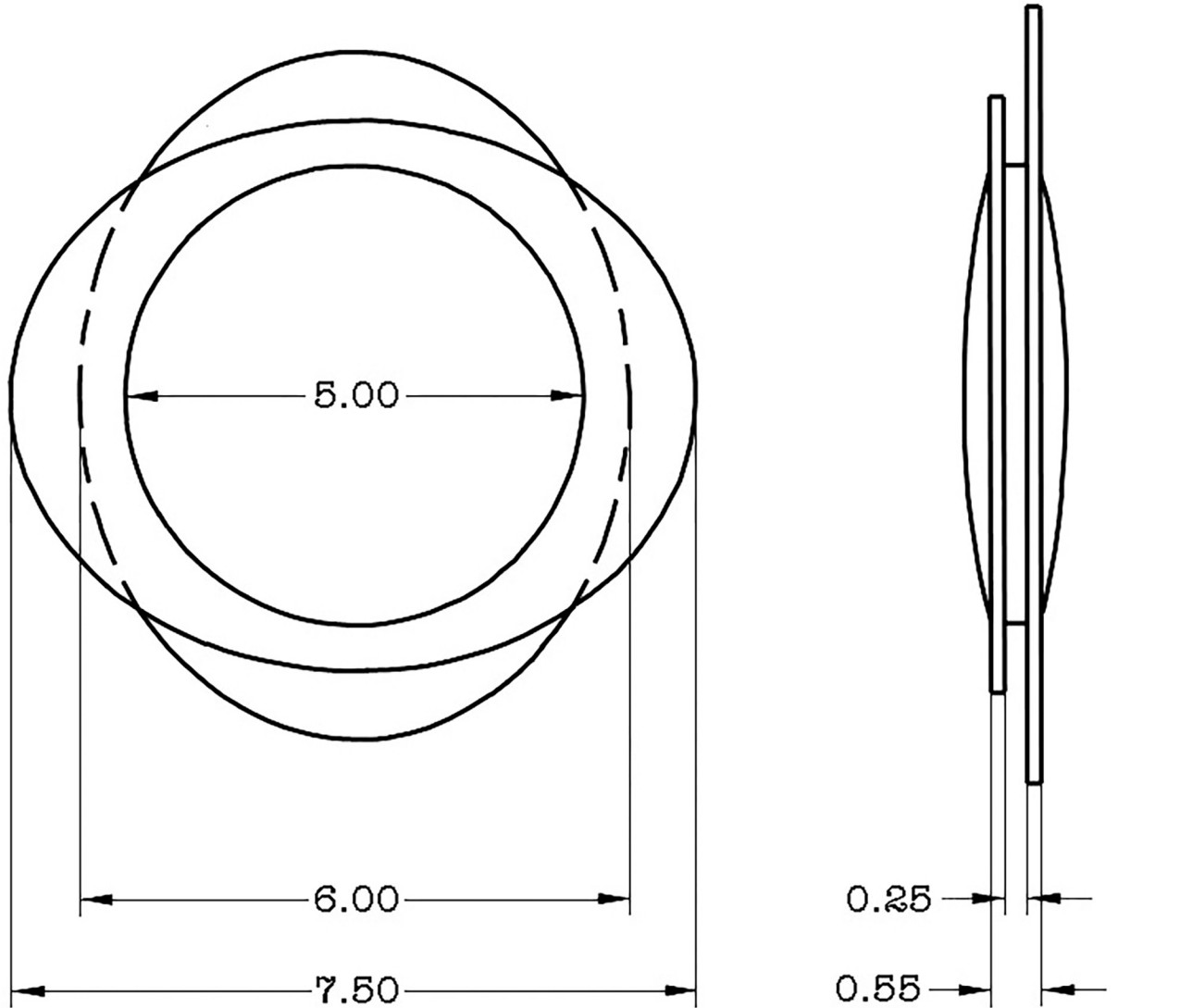

**Fig 1. Technical drawing of the 'bag-in-the-lens' (BIL) biconvex design, Morcher 98A, as published in: US patent 6,027,531 on February the 22nd, 2000.** The patents belong to the public domain since 2018. The figure was illustrated by Marie-José Tassignon.

phacoemulsification. Despite these advances however, PCME remains an important cause of decreased visual acuity (VA) after cataract surgery. Systemic and ocular comorbidities known to be risk factors for PCME are: older age, males, preoperative use of prostaglandins, uveitis, retinal vein occlusion, a preexisting epiretinal membrane, or retinal detachment repair [19–22]. Patients with diabetes, even in the absence of diabetic retinopathy, have an increased risk for developing macular edema postoperatively. The risk is even higher in the presence of diabetic retinopathy and increases proportionally based on the severity of the retinopathy [23,24]. Surgical complications that influence PCME development include iris trauma, vitreous traction, vitreous prolapse and loss, vitrectomy, IOL dislocation, early postoperative capsulotomy, posterior capsule rupture, and the use of iris-fixated or anterior chamber IOLs [5,25].

Given the extra surgical steps required by the BIL implantation technique, it is important that the non-inferiority approach be validated with respect to postoperative complications

such as PCME. In this study, we aimed to examine the total incidence of clinically significant pseudophakic macular edema (CSPME) after BIL implantation in a large cohort of adult patients and examine non-inferiority with comparison to the traditional LIB surgical technique. The term 'clinically significant' is used because postoperative optical coherence tomography (OCT) was not performed routinely in this retrospective study and was only implemented when the treating physician requested it. The secondary aim was to study risk factors for CSPME and assess incidence in patients with and without risk factors. This was to define a difference between postoperative significant pseudophakic macular edema (CSPME) caused by inflammation of the surgery, and clinically significant cystoid macular edema (CSME) caused by structural abnormalities.

In CSPME, the main etiology is thought to be the upregulation of inflammatory mediators in the aqueous and vitreous humor after surgical manipulation, which breaks down the blood-aqueous and blood-retinal barriers causing increased vascular permeability [25]. Eosinophilic transudates accumulate in the outer plexiform and inner nuclear layers of the retina to create cystic spaces that coalesce to form larger pockets of fluid. This type of macular edema responds well to systemic or topical anti-inflammatory therapy. In CSME, the leakage and edema is mediated by structural abnormalities in retinal vasculature, like in diabetic retinopathy and exudative macular degeneration, or by vitreomacular traction. These patients require treatment with intravitreal antivascular endothelial growth factor therapy or macular pucker or internal limiting membrane peeling.

## Materials and methods

This retrospective study was a single-center, observational cohort study with a longitudinal design. The data were derived from the patient population in a university hospital setting. The Ethics Committee of Antwerp University Hospital and the University of Antwerp (17/18/229) approved the research and waived the need of informed consent since it concerns a retrospective study. The study and analysis were executed according to the tenets of the Declaration of Helsinki. This study complements a previous study with an extension of the amount of data analyzed [5].

### Data collection

All surgeries that took place during a 6-year period, starting from January 2013 and ending on December 2018, were considered for inclusion. The first-operated eye of each BIL implantation was included in the study. The data included the following parameters for each patient: age at the time of surgery, sex, operated eye side, operation date. Patient demographic data, medical history, intraoperative and postoperative complications were retrieved from the electronic medical registers. Exclusion factors were: patients under 18 years of age, IOL exchanges, combined procedures and cases who had ocular surgery within the 3 months prior to the cataract surgery. Patients who suffered from CME due to diabetes or other causes were included in the study provided they were stable and at their CME baseline on OCT for at least three months prior to the cataract surgery. The data was collected in an anonymized database, which was uploaded with the submission of this article, except for 'date of birth' and 'age' because this information could compromise patients' anonymity.

### BIL implantation technique

We describe the outcomes of six different surgeons with excellent experience in the practice of the BIL technique (co-authors LVO, SND, MJT, SK, JVL, VDG). The procedures were all performed under topical anesthesia, a temporal limbal incision, an intracameral injection of

adrenaline/lidocaine 1: 1000, filling of the anterior chamber with an ophthalmic viscosurgical device (OVD) (Healon GV–Johnson & Johnson, New Brunswick, NJ, USA) and a centered, calibrated 4.8mm to maximum 5mm capsulorhexis of the anterior capsule. For this purpose, the ring caliper of 5 mm (Morcher 5L) was used as previously described. The cataract lens was removed using phacoemulsification, and a PPCCC of an approximately equal size to the anterior capsulorhexis, was performed. The anterior and posterior rhexis were inserted into the equatorial groove of the BIL. Miostat 0.01% (Carbachol ®, Alcon Laboratories, Fort Worth, Texas, USA) was injected intracamerally to induce miosis and prevent immediate postoperative iris incarceration. Intracameral cefuroxime prophylaxis (Aprokam ®, Théa Pharma, Haarlem, The Netherlands) was used as standard at the end of surgery.

## Postoperative follow-up

All patients received topical diclofenac (Dicloabak®, Théa Pharma, Haarlem, The Netherlands) 4 times a day for 5 weeks postoperatively together with a topical suspension of tobramycin combined with dexamethasone (Tobradex®, Novartis, Bazel, Switzerland) 4 times a day for 1 week, after which it was either stopped in the absence of inflammation or tapered by 1 drop a week over 4 weeks in case of persisting inflammation. Postoperative examinations were performed at 1 day, 1 week, and 5 weeks. The patients were also informed that in case of a visual decline, they should contact the ophthalmic emergency department to exclude complications. Next examinations were at 6 months and one year postoperatively. The corrected distance visual acuity (CDVA) and retinal status was evaluated during these consecutive checks. When CDVA was 10/10 on the 5 weeks postoperatively visit, the patient was requested to come only in case of decreased visual acuity.

It was expected that CSPME would occur before the 3-month postoperative interval. Spectral-domain OCT (SD-OCT) was only performed on clinical indication: in case decline of CDVA was not explained by corneal edema, macular pathologies or optic disc pathologies or in case of presentation with metamorphopsia, scotoma, micropsia, and/or suspicious fundoscopic findings. CSPME was diagnosed on OCT if there was (1) the presence of new or deterioration of preoperative recorded cystic changes within the neurosensory retina or subretinal fluid; (2) an increase of greater than or equal to 50 μm in central retinal thickness compared with the lowest previous measurement; and (3) any increase in central retinal thickness, in conjunction with a loss of more than 5 letters from the best previous postoperative measurement. In the absence of a preoperative SD-OCT, measurements above the SDs of 259.7 μ (±21.5) in men and 250.1 μ (±21.6) in women were diagnosed as increased retinal thickness, taken the clinical context into account.

## Statistical analysis

The data were analyzed using the statistical program, R (version 3.6.1, R Foundation, Vienna, Austria). Baseline characteristics were reported as number (percentage) for categorical variables and as mean (SD) or median (range) for continuous variables. The follow-up was limited to 6 months. The 3-month incidence rate of CSME and CSPME was computed using the Kaplan-Meier method. Because there is evidence from the literature and clinic that CSPME develops between 10 days and 6 weeks postoperatively, the 3-month cumulative incidence was considered the most informative summary. To show non-inferiority, the upper boundary of the 95% confidence interval of the 3-month incidence rate is compared to the a priori defined non-inferiority margin. This margin was set at the upper range of CSPME in patients with risk factors after LIB implantation reported in literature (2.35%). If the upper boundary of the 95% CI for the 3-month cumulative incidence rate of CSPME falls below this margin, non-inferiority can be concluded.

Additionally, a non-inferiority test was performed as a one-sided log-rank test showing the incidence is significantly less than the reported incidence of 2.35% in literature [22,27].

The impact of the previously described risk factors on the development of CSME or CSPME after the BIL implantation technique was studied with Cox proportional hazards regression. In this way data from patients with short follow-up can be taken into account as censored observations. First, simple models with only 1 risk factor were created. All factors with a P value less than 0.2 were elected for inclusion in a multiple Cox model, and a stepwise construction ended up in a concluding model in which all P values were below the significance level of .05. Because of the small number of events, no more than 3 independent variables were included in the final model. Hazard ratios (HRs), and corresponding 95% CIs were reported.

A minimum of 1285 patients needed to be included in the study to show non-inferiority with 80% power, confidence level 95% and non-inferiority limit of 2.35%. We hereby assumed an incidence rate of 1.4% based on previous research [5].

## Results

Of all the individuals who underwent BIL cataract surgery during this 6-year study period, 2419 patients met the inclusion criteria. Their clinical characteristics are presented in Table 1. Of this group, 1061 (44%) were men and 1358 (56%) were women. 1236 patients (about 50%) were followed up at least 3 months postoperatively.

The mean age was 70 years (SD ±13), and 404 patients (17%) were younger than 60 years. The median time to develop CSPME was 34 days (ranging from 6 to 84 days). A detailed breakdown of the 15 patients who developed clinically significant cystoid macular edema after BIL cataract surgery with their characteristics and risk factors is shown in Table 2A (CSPME: n = 5) and 2B (CSME: n = 10) respectively.

The 3-month incidence of CSPME in the subgroup without risk factors was 0.00% (95%CI: 0.00 –NA), 0.57% (95% CI 0.22–1.29%) in the subgroup with risk factors and 0.29% (95% CI: 0.11–0.65%) in the total population. The 3-month incidence of CSME in the subgroup without risk factors was 0.00% (0.00-NA), 1.18% (95% CI: 0. 61–2.10%) in the subgroup with risk factors and 0.59% (95% CI: 0.30–1.07%) in the total population. The 3-month incidence of CSME or CSPME in the subgroup without risk factors was 0.00% (95% CI: 0.00 –NA), 1.75% (95% CI: 1.02–2.81%) in the subgroup with risk factors and 0.87% (95% CI: 0.51–1.42%) in the total population. A cumulative incidence curve of the events of CSME and CSPME, including 95% CI and number at risk, was made to visualize the time to event (Fig 2).

The upper boundary of the confidence interval of the 3-month cumulative incidence of CSPME in the group with risk factors (1.29%) is lower than the non-inferiority margin, namely the upper range of CSPME in the group with risk factors after the LIB implantation, reported in literature (2.35%) [22,28]. Therefore it can be concluded that the BIL implantation technique is non-inferior to the LIB implantation technique regarding the development of CSPME in this vulnerable group (P < 0.001).

The risk factors examined (Table 1), were divided into ocular and systemic variables. The simple Cox regression models revealed a highly significant impact of renal insufficiency (HR: 7.93 [95% CI: 2.56–24.59, P = .003]), age related macular edema (HR: 7.1 [95% CI: 2.58–19.56, P < .001]) and more specifically the age related exudative macular edema (HR: 64.59 [95% CI: 23.47–177.77, P < .001]). Furthermore, retinal venous occlusion (HR: 18.95 [95% CI: 4.31–83.42, P .005]), vitreomacular traction (HR: 12.04 [95% CI: 4.18–34.71, P < .001]), intravitreal injections (HR: 18.17 [95% CI: 5.17–63.84, P < .001]) and ipsilateral CSME (HR: 22.68 [95% CI: 7.31–70.4, P < .001]) were significantly associated with CSPME. There were too few events with history of dry AMD, prostaglandin use preoperatively, retinal defect, intraocular surgery,

**Table 1. Patient characteristics.**

|  | Total population (N = 2419) | At least 90 d FU (N = 1236) | CSME (N = 10) | CSPME (N = 5) | CSME or CSPME (N = 15) |
|---|---|---|---|---|---|
| **Age (y)** |  |  |  |  |  |
| Mean (SD) | 70 (13) | 68 (13) | 73 (10) | 72 (1.3) | 73 (8) |
| Median (min–max) | 72 (8–120) | 71 (10–120) | 74 (57–84) | 73 (71–74) | 73 (57–84) |
| < 60 y | 404 (17%) | 240 (19%) | 2 (20%) | 0 (0%) | 2 (13%) |
| **Gender (male)** |  |  |  |  |  |
| Female | 1358 (56%) | 699 (57%) | 5 (50%) | 2 (40%) | 7 (47%) |
| Male | 1061 (44%) | 537 (43%) | 5 (50%) | 3 (60%) | 8 (53%) |
| **History** |  |  |  |  |  |
| [a]Renal insufficiency | 84 (3%) | 56 (5%) | 1 (10%) | 3 (60%) | 4 (27%) |
| Diabetes mellitus | 294 (12%) | 190 (15%) | 2 (20%) | 2 (40%) | 4 (27%) |
| DR | 85 (4%) | 67 (5%) | 1 (10%) | 0 (0%) | 1 (7%) |
| AMD | 149 (6%) | 118 (10%) | 6 (60%) | 0 (0%) | 6 (40%) |
| AMD (dry) | 129 (5%) | 101 (8%) | 0 (0%) | 0 (0%) | 0 (0%) |
| AMD (exudative) | 20 (1%) | 17 (1%) | 6 (60%) | 0 (0%) | 6 (40%) |
| Preoperative prostaglandin use | 104 (4%) | 75 (6%) | 0 (0%) | 0 (0%) | 0 (0%) |
| Uveïtis | 49 (2%) | 39 (3%) | 3 (30%) | 0 (0%) | 3 (20%) |
| RVO | 18 (1%) | 11 (1%) | 1 (10%) | 1 (20%) | 2 (13%) |
| VMT | 71 (3%) | 58 (5%) | 3 (30%) | 2 (40%) | 5 (33%) |
| Myopic maculopathy | 14 (1%) | 13 (1%) | 1 (10%) | 0 (0%) | 1 (7%) |
| High myopia | 335 (14%) | 198 (16%) | 1 (10%) | 0 (0%) | 1 (7%) |
| Retinal defect | 53 (2%) | 39 (3%) | 0 (0%) | 0 (0%) | 0 (0%) |
| Intraocular surgery | 80 (3%) | 66 (5%) | 0 (0%) | 0 (0%) | 0 (0%) |
| Intravitreal injections | 23 (1%) | 21 (2%) | 3 (30%) | 0 (0%) | 3 (20%) |
| Ipsilateral CSME | 31 (1%) | 25 (2%) | 4 (40%) | 0 (0%) | 4 (27%) |
| Contralateral CSME | 35 (1%) | 23 (2%) | 1 (10%) | 0 (0%) | 1 (7%) |
| Contralateral PCME | 4 (0%) | 3 (0%) | 0 (0%) | 0 (0%) | 0 (0%) |
| Difficult mydriasis | 216 (9%) | 120 (10%) | 0 (0%) | 1 (20%) | 1 (7%) |
| **Complications** |  |  |  |  |  |
| Peroperative | 47 (2%) | 27 (2%) | 0 (0%) | 0 (0%) | 0 (0%) |
| Postoperative | 29 (1%) | 16 (1%) | 0 (0%) | 0 (0%) | 0 (0%) |
| **Time to event/last follow-up (d)** |  |  |  |  |  |
| Median (min–max) | 95 (0–1590) | 394 (6–1590) | 35 (6–84) | 27 (6–61) | 34 (6–84) |

N = number of patients; FU = follow-up; d = days, y = years; AMD = age-related macular degeneration; CSME = clinically significant macular edema;

CSPME = clinically significant pseudophakic cystoid macular edema; DR = diabetic retinopathy; RVO = retinal vein occlusion; VMT = vitreomacular traction;

[a]Glomerular filtration rate according to the 'Modification of Diet in Renal Disease' study < 60 mL/min/1.73 m$^2$.

contralateral PCME, peroperative or postoperative complication to fit a Cox model. Three risk factors were confirmed in the final multiple Cox regression analysis. The development of CSME or CSPME was independently associated with renal insufficiency (HR: 5.42; 95% CI: 1.69–17.44; P = .014), exudative age-related macular degeneration (HR: 74.50, 95% CI: 25.75–215.6; P < .001) and retinal vein occlusion (HR: 22.48, 95% CI: 4.55–111.02; P = .005) (Table 3).

## Discussion

CSPME is a known complication of cataract surgery which occurs more frequently in patients with risk factors. However, examining the rate of edema development of a new surgical

**Table 2. a. Patient characteristics of CSPME after the BIL implantation technique.** b. Patient characteristics of CSME after the BIL implantation technique.

| N° | Time to CSPME (d) | Sex | Age (y) | History of risk factors |
|---|---|---|---|---|
| 1 | 34 | Female | 71 | Chronic renal insufficiency, diabetes mellitus type 2 |
| 2 | 7 | Female | 73 | Discrete ERM (Juxtafoveal cyst resolved after treatment with Acetazolamide PO) |
| 3 | 28 | Male | 71 | Chronic renal insufficiency (nefroangiosclerosis), difficult mydriasis (malyugin ring) |
| 4 | 62 | Male | 74 | Chronic renal insufficiency, diabetes mellitus, VMT (discrete ERM) |
| 5 | 5 | Male | 73 | History of, CRVO (2 years preoperatively, no CSME preoperatively) |

| N° | Time to CSME (d) | Sex | Age (y) | History of risk factors |
|---|---|---|---|---|
| 1 | 83 | Female | 79 | Exudative AMD, acute renal failure (angiosclerosis) |
| 2 | 6 | Male | 57 | Fuchs' uveitis, VMT (CSME resolved after pucker peeling) |
| 3 | 22 | Male | 75 | Diabetes mellitus type 2, exudative AMD |
| 4 | 44 | Female | 67 | CRVO |
| 5 | 8 | Female | 57 | VMT, ipsilateral CSME, high myopia, myopic maculopathy, intravitreal injections |
| 6 | 42 | Male | 73 | Exudative AMD |
| 7 | 37 | Female | 75 | Dry AMD ipsilateral, HLA B27–associated uveitis, diabetes mellitus, prostaglandin use preoperatively, VMT (pucker), ipsilateral and contralateral CSME |
| 8 | 7 | Male | 73 | Exudative AMD, history of ipsilateral CSME, intravitreal injections |
| 9 | 34 | Female | 82 | Exudative DR, herpetic uveitis, diabetes mellitus |
| 10 | 42 | Female | 84 | Exudative AMD, ipsilateral CSME, intravitreal injections |

AMD = age-related macular degeneration; BIL = bag-in-the-lens; CSME = clinically significant macular edema; CRVO = central retinal vein occlusion; CSPME = clinically significant pseudophakic cystoid macular edema; DR = diabetic retinopathy; HLA = human leukocyte antigen; VMT = vitreomacular traction.

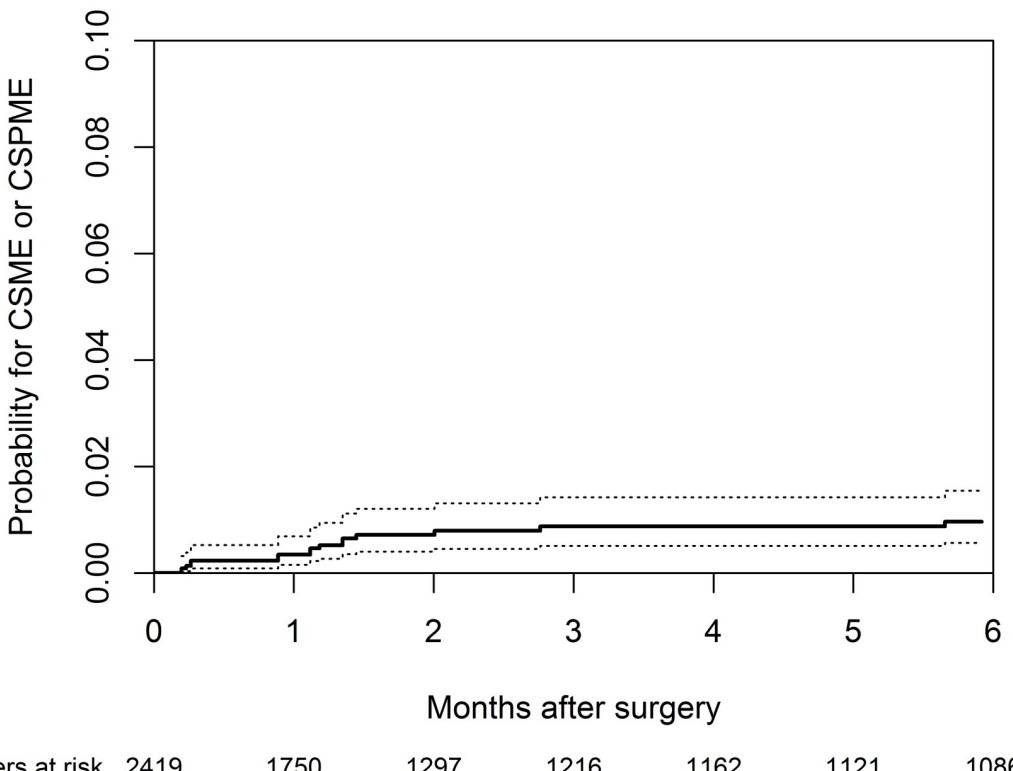

Numbers at risk 2419 1750 1297 1216 1162 1121 1086

**Fig 2. The cumulative incidence of Clinically Significant Cystoid Macular Edema (CSME) caused by risk factors and Clinically Significant Pseudophakic Cystoid Macular Edema (CSPME) after 'after the 'bag-in-the-lens' (BIL) implantation.**

**Table 3. Simple and multiple Cox models for CSME and CSPME after BIL implantation.**

| Patient features | Simple Cox models | | Multiple Cox model | |
|---|---|---|---|---|
| | HR (95%CI) | P (LR-test) | HR (95%CI) | P (LR-test) |
| **Characteristics** | 1.02 (0.98,1.07) | | | |
| Age (years) | 1.02 (0.98,1.07) | 0.29 | | |
| < 60 years of age | 0.64 (0.14, 2.81) | 0.13 | | |
| Gender (male) | 1.62 (0.6,4.34) | 0.34 | | |
| **Systemic risk factors** | | | | |
| Renal insufficiency | 7.93 (2.56,24.59) | 0.003 | 5.42 (1.69, 17.44) | 0.014 |
| Diabetes mellitus | 2.06 (0.66,6.39) | 0.24 | | |
| **Ophthalmic risk factors** | | | | |
| DR | 1.41 (0.19,10.71) | 0.75 | | |
| AMD | 7.1 (2.58,19.56) | < 0.001 | | |
| AMD (exudative) | 64.59 (23.47,177.77) | < 0.001 | 74.50 (25.75, 215.6) | < 0.001 |
| Uveïtis | 9.14 (2.6,32.12) | 0.006 | | |
| RVO | 18.95 (4.31,83.42) | 0.005 | 22.48 (4.55, 111.02) | 0.005 |
| VMT | 12.04 (4.18,34.71) | < 0.001 | | |
| Myopic maculopathy | 8.56 (1.13,64.89) | 0.12 | | |
| High myopia | 0.37 (0.05,2.84) | 0.27 | | |
| Intravitreal injections | 18.17 (5.17,63.84) | < 0.001 | | |
| Ipsilateral CSME | 22.68 (7.31,70.4) | < 0.001 | | |
| Contralateral CSME | 4.09 (0.54,31.01) | 0.26 | | |
| Difficult mydriasis | 1.33 (0.3,5.84) | 0.72 | | |

AMD = age-relatedmacular degeneration; BIL = bag-in-the-lens; CSME = clinically significant macular edema; CSPME = clinically significant pseudophakic cystoid macular edema; DR = diabetic retinopathy; HR = hazard ratio; LR = likelihood-ratio; RVO = retinal vein occlusion; VMT = vitreomacular traction.

technique can be difficult because of the relative rarity of CSPME in addition to the myriad of risk factors associated with its development. These can easily confound the true rate of incidence associated with the lens technique. CSPME can develop subclinically, being visible on imaging techniques, such as OCT or fluorescein angiography, but without causing symptoms. In patients with risk factors or after complicated surgery, the clinical reports describe varying from 0.1% to 2.35%, whereas imaging-based examinations report rates from 4% to 60% [22,26].

The retrospective nature of this study, even if conducted in a large cohort of patients, remains a major limitation in drawing conclusions. It meant that the CSPME rate was documented on the basis of clinical examination and OCT on request in case of decreased visual acuity [27]. As subclinical cystoid macular edema was not taken into account in this study, the output cannot be correlated with the results documented by Wielders et al [28]. Patients were sub-analyzed with and without risk factors to try to segregate the impact from the IOL technique itself from bias. It is also important to consider the sample size. Although the study group is large, CSPME has a low incidence and analyses are based on only 15 cases. Some of the risk factors were rare, e.g. a history of uveitis, retinal venous occlusion, exudative age-related macular degeneration, prostaglandin use, retinal defects, intraocular surgery and intra- and postoperative complications. This implies convergence problems in the models. The results should be interpreted with care, even for the risk factors for which a model could be fitted, especially when extremely large HRs and wide CIs are found, demonstrating the ambiguity of the model projections.

Moreover, not all possible risk factors that are cited in the literature could be confirmed decisively in this cohort. It is important to consider that all patients received topical nonsteroidal anti-inflammatory drops postoperatively, which can influence the CSPME incidence. However, this treatment regimen reflects current clinical practice in most centers. To determine the exact incidence of CSPME in the presence of risk factors, future multicenter prospective research is indispensable. Defining these risk factors assists in optimizing pre- and postoperative care. In the absence of risk factors, the rate of CSPME after BIL cataract surgery was zero. The presence of risk factors increased the incidence of CSPME, as expected. This study proves that patients with renal insufficiency have an independently higher risk to develop CSME/CSPME. The hypothesis for this association is that there are similarities in the structure and function of the basement membrane of the nephrotic glomeruli and the ocular choroidea and that dysfunction of these anatomical structures leads to fluid leakage.

In conclusion this case series study suggested the non-inferiority of BIL surgery regarding the development of CSME or CSPME, relative to the traditional 'lens-in-the-bag' (LIB) implantation technique, in the first 3 months postoperative. We attribute this to the fact that the opening of the posterior capsule is performed in a highly controlled manner, preserving the integrity of the anterior hyaloid that functions as the main barrier between anterior and posterior segment.

## Supporting information

**S1 Dataset.**
(XLSX)

## Author Contributions

**Conceptualization:** Marie-José Tassignon.

**Data curation:** Jasmien Rens.

**Formal analysis:** Dorothée Scheers, Kristien Wouters.

**Investigation:** Dorothée Scheers.

**Methodology:** Dorothée Scheers, Marie-José Tassignon.

**Supervision:** Luc Van Os, Sorcha Ní Dhubhghaill, Veva De Groot, Stefan Kiekens, Jan Van Looveren, Marie-José Tassignon.

**Writing – original draft:** Dorothée Scheers.

**Writing – review & editing:** Jasmien Rens, Marie-José Tassignon.

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
