## [Decision Letter · Decision Letter 0]

19 Sep 2022

PONE-D-22-20385Safety of the bag-in-the-lens implantation regarding the development of clinically significant pseudophakic cystoid macular edema: a retrospective case series studyPLOS ONE

Dear Dr. Rens,

Thank you for submitting your manuscript to PLOS ONE. After careful consideration, we feel that it has merit but does not fully meet PLOS ONE’s publication criteria as it currently stands. Therefore, we invite you to submit a revised version of the manuscript that addresses the points raised during the review process.

We look forward to receiving your revised manuscript.

Kind regards,

Prof. Andrzej Grzybowski, MD, PhD, MAE, MBA

Academic Editor

PLOS ONE

Journal Requirements:

Reviewers' comments:

Reviewer's Responses to Questions

**Comments to the Author**

1. Is the manuscript technically sound, and do the data support the conclusions?

Reviewer #1: Yes

Reviewer #2: Yes

2. Has the statistical analysis been performed appropriately and rigorously? 

Reviewer #1: Yes

Reviewer #2: Yes

3. Have the authors made all data underlying the findings in their manuscript fully available?

Reviewer #1: Yes

Reviewer #2: Yes

4. Is the manuscript presented in an intelligible fashion and written in standard English?

Reviewer #1: Yes

Reviewer #2: Yes

5. Review Comments to the Author

Reviewer #1: This a relevant manuscript for those performing are planning on performing BIL surgery. The manuscript is sound and clear. The statistical analysis is ok. I would focus more on odds ratios additionally to the already used HR in the results section.

I would recommend to show the difference in central macular thickness (pre- to post-op) as a box plot.

Reviewer #2: The authors retrospectively reviewed records of 2400 patients who underwent the "bag in the lens" technique and reported CME prevalence in patients with and without predisposing risk factors.

The study is well designed but the main draw back is retrospective nature of the study which makes us to conclude with caution.

Retrospective study has many limitations and It will be defective in CME evaluation and recording specially when we have no OCT.

I suggest adding a limitation part ad mentioning all of these in this part. CME in an important concern in Post capsulorhexis and I think this study can not prove that there is no significant difference in CME with and without post rhexis .

6. PLOS authors have the option to publish the peer review history of their article (what does this mean?). If published, this will include your full peer review and any attached files.

Reviewer #1: No

Reviewer #2: **Yes: **Hesam Hashemian

---

## [Author Response · Author response to Decision Letter 0]

5 Nov 2022

Our response to the reviewers can also be found in the rebuttal letter. 

Thank you for your time in reviewing our paper in depth. Please find our reply to your comments:

- Adding odds ratios would mean adding a logistic regression model for the event rate over a certain period (e.g. 3 month). However, as not all patients had the same follow-up term, a Cox proportional hazards model was considered to be more appropriate to study the impact of risk factors on the occurrence of CSME/CSPME. In this way, patients with less than 3 months follow-up are not lost, as would be the case in a logistic regression model for the event rate at 3 months. The Cox proportional hazards model is summarized in hazard ratios instead of odds ratios, however interpretation is similar. The choice of the Cox model is further clarified in the paper. 

- The central retinal thickness was not included as a parameter in our statistical analysis. We only used the central retinal thickness in case of decline in CDVA to confirm the diagnosis of CSPME. As a result, pre- or post-op central macular thickness was not a parameter included in this paper.

---

## [Decision Letter · Decision Letter 1]

28 Nov 2022

Safety of the bag-in-the-lens implantation regarding the development of clinically significant pseudophakic cystoid macular edema: a retrospective case series study

PONE-D-22-20385R1

Dear Dr. Rens,

We’re pleased to inform you that your manuscript has been judged scientifically suitable for publication and will be formally accepted for publication once it meets all outstanding technical requirements.

Kind regards,

Andrzej Grzybowski

Academic Editor

PLOS ONE

**Comments to the Author**

1. If the authors have adequately addressed your comments raised in a previous round of review and you feel that this manuscript is now acceptable for publication, you may indicate that here to bypass the “Comments to the Author” section, enter your conflict of interest statement in the “Confidential to Editor” section, and submit your "Accept" recommendation.

Reviewer #1: All comments have been addressed

Reviewer #2: All comments have been addressed

2. Is the manuscript technically sound, and do the data support the conclusions?

Reviewer #1: Yes

Reviewer #2: Yes

3. Has the statistical analysis been performed appropriately and rigorously? 

Reviewer #1: Yes

Reviewer #2: Yes

4. Have the authors made all data underlying the findings in their manuscript fully available?

Reviewer #1: Yes

Reviewer #2: Yes

5. Is the manuscript presented in an intelligible fashion and written in standard English?

Reviewer #1: Yes

Reviewer #2: Yes

6. Review Comments to the Author

Reviewer #1: The authors replied to all suggestions. Only minor changes were performed. There are no further suggestions from my side.

Reviewer #2: I believe that retrospective study can not fully evaluate clinical Macular edema and the power of the study is not good.

7. PLOS authors have the option to publish the peer review history of their article (what does this mean?). If published, this will include your full peer review and any attached files.

Reviewer #1: No

Reviewer #2: No

---

## [Editor Report · Acceptance letter]

28 Dec 2022

PONE-D-22-20385R1 

Safety of the bag-in-the-lens implantation regarding the development of clinically significant pseudophakic cystoid macular edema: a retrospective case series study 

Dear Dr. Rens:

I'm pleased to inform you that your manuscript has been deemed suitable for publication in PLOS ONE. Congratulations! Your manuscript is now with our production department. 

Kind regards, 

on behalf of

Dr. Andrzej Grzybowski 

Academic Editor

PLOS ONE